# Using Pathogenic *Escherichia coli* Type III Secreted Effectors *espK* and *espV* as Markers to Reduce the Risk of Potentially Enterohemorrhagic Shiga Toxin-Producing *Escherichia coli* in Beef

**DOI:** 10.3390/foods14030382

**Published:** 2025-01-24

**Authors:** Joseph M. Bosilevac, Tatum S. Katz, Leslie E. Manis, Lorenza Rozier, Michael Day

**Affiliations:** 1Meat Safety and Quality Research Unit, Roman L. Hruska United States Meat Animal Research Center, Agricultural Research Service, United States Department of Agriculture, Clay Center, NE 68933, USA; tatum.katz@usda.gov; 2Office of Public Health Science Midwest Laboratory, Food Safety and Inspection Service, United States Department of Agriculture, St. Louis, MO 63310, USA; lesliedk2@yahoo.com; 3Office of Public Health Science Eastern Laboratory, Russell Research Center, Food Safety and Inspection Service, United States Department of Agriculture, Athens, GA 30605, USA; lorenza.rozier@usda.gov (L.R.); michael.day@usda.gov (M.D.)

**Keywords:** Shiga toxin-producing *Escherichia coli*, virulence factors, molecular detection, beef, risk reduction

## Abstract

Contamination of beef by certain strains of Shiga toxin-producing *Escherichia coli* (STEC) called enterohemorrhagic *E. coli* (EHEC) can lead to outbreaks of severe disease. Therefore, accurate monitoring tests are needed to identify high risk beef products and divert them from consumers. Most EHEC testing focuses on the detection of their key virulence factors Shiga toxin (*stx*) and intimin (*eae).* However, these two factors can occur separately in lower risk nonpathogenic *E. coli* (STEC and enteropathogenic *E. coli*; EPEC) and confound testing if both are present. Accessory virulence factors like the Type III secreted effectors *espK* and *espV* may aid in increasing the specificity of EHEC testing. This work first evaluated collections of EHEC (n = 83), STEC (n = 100) and EPEC (n = 95), finding *espK* and/or *espV* in 100%, 0%, and 60% of each, respectively. Next, an inoculation study of beef trim samples (n = 118) examined the ability of including *espK* and *espV* in the monitoring test scheme to distinguish samples inoculated with EHEC from those inoculated with mixtures of STEC and EPEC (non-EHEC). Test accuracy was calculated as Area Under the Receiver Operating Characteristic curve (AUC) and found to be significantly (*p* < 0.05) different, increasing from 68.0% (*stx*/*eae*) to 76.8% by including *espK* and *espV*. Finally, 361 regulatory agency beef samples that had been identified as suspect for EHEC (*stx*+/*eae*+) were examined with the addition of *espK* and *espV*, and results compared to culture isolation. Culture isolation identified 42 EHEC, 82 STEC, and 67 EPEC isolates in 146 of the samples. In the case of these naturally contaminated samples, inclusion of *espK* and *espV* increased test accuracy compared to culture isolation from an AUC of 50.5% (random agreement) to 69.8% (good agreement). Results show that the inclusion of *espK* and *espV* can increase the specificity of identifying high risk EHEC contaminated beef and release beef contaminated with nonpathogenic or low risk *E. coli*. Further, use of *espK* and *espV* identified samples contaminated by common EHEC of serogroups O157, O26, and O103, as well as of less common serogroups O182, O177, and O5.

## 1. Introduction

Shiga toxin-producing *Escherichia coli* (STEC) are a cause of foodborne infections that range from uncomplicated diarrhea to life-threatening conditions like hemorrhagic colitis or hemolytic-uremic syndrome. The STEC strains associated with severe clinical illness and enterohemorrhagic diseases can also be referred to as enterohemorrhagic *E. coli* (EHEC). EHEC carry additional virulence factors, such as intimin (*eae*, a gene encoded by the locus of enterocyte effacement [LEE]). Intimin is involved in the formation of attaching and effacing (A/E) lesions on enterocytes during infection of the human gut [1]. The LEE also encodes a type III secretion system (TTSS), regulatory elements, as well as secreted effector proteins and their chaperones [2]. Additional non-LEE located effector (*nle*) proteins associated with pathogenicity are secreted as well. Many *nle* proteins are expressed from integrated phage or pathogenicity islands, termed O-islands (OI; [3]).

Two such nle factors are *espK* and *espV*. *espK* is a gene located on OI-50, and was shown to be a locus involved in the persistence of EHEC O157:H7 in the intestines of orally inoculated calves [4]). *espV* is an avirulence A (AvrA) family effector and was revealed by genome sequencing to be a common genetic marker located on OI-44 in different severe disease-causing EHEC [5]. The role of *espV* in disease is not known; however, it was shown to cause drastic morphological changes (nuclear condensation, cell rounding, and dendrite-like projections) when expressed in mammalian cells [6]. *espK* and *espV* were found to be more prevalent in EHEC compared to enteropathogenic *E. coli* (EPEC) which possess the prototype LEE but are not STEC. *espK* was found present in 92.4% of EHEC and 28.8% of EPEC, and *espV* was found in 84.4% of EHEC and 45.2% of EPEC when 1100 strains isolated from humans, animals, and food sources were examined [7]. When *espK* and *espV* were analyzed as a combination (*espK* and/or *espV*), 98.5% of EHEC and 54.1% of EPEC were found positive, while only 1.6% of STEC and 1.1% of nonpathogenic *E. coli* were found positive [7]. Therefore, *espK* and *espV* appear to be good candidates as genetic markers for discriminating EHEC from non-EHEC strains.

Among strains of EHEC, seven serogroups (O26, O45, O103, O111, O121, O145, and O157) are associated with severe clinical illness in humans [8]. The development of methods for their reliable detection from food has been challenging. This is especially so because the EHEC can be present at low levels, requiring the food sample to be culturally enriched. Therefore, PCR detection of the EHEC virulence genes *stx*_1_, *stx*_2_, *eae*, and O-serogroup-specific genes is useful but does not identify EHEC strains specifically since other *E. coli* (STEC and EPEC) may possess them. Currently, standard methods using *stx*, *eae*, and O-serogroup-specific genes for detecting the top seven EHEC serogroups result in numerous samples identified as potential positives, but where an EHEC cannot be culturally confirmed. By their own estimate, the United States Department of Agriculture (USDA) Food Safety and Inspection Service (FSIS) has stated that approximately 80% of non-O157 EHEC screen-positive samples cannot be confirmed [9]. The addition of *espK* and *espV* to the screening approach was shown to narrow down 180 (*stx*+/*eae*+) implicated beef enrichments to 80, making the EHEC detection more reliable [10].

Recently, a commercial assay for pathogenic *E. coli* (PEC) that targets *espK* and *espV* was released. The goal of this work was to evaluate the PEC assay in combination with *stx* and *eae* screening to determine its influence on increasing the specificity (i.e., fewer false positives) of EHEC detection. The PEC assay was first used to examine a collection of EHEC, EPEC, and STEC isolates. Then its ability to discriminate mixed cultures of EPEC and STEC from EHEC inoculated beef was examined. Finally, the new assay was used to screen regulatory beef enrichment broths that had been cultured for EHEC and determine what impact it would have on routine EHEC testing of beef products.

## 2. Materials and Methods

### 2.1. Molecular Screening Assays

The template for molecular detection was generated using GENE-UP^®^ DNA lysis kits (bioMerieux Inc., Hazelwood, MO, USA). Tests for *stx* and *eae* and pathogenic *E. coli* (*espK*/*espV*) were performed using EH1^®^ and PEC^®^ test kits (bioMerieux Inc.), respectively. Further, testing for *E. coli* O157:H7 and non-O157 STEC serogroups (O26, O45, O103, O111, O121, and O145) was performed using bioMerieux ECO^®^ and EH2^®^ test kits, respectively. All test kits were used according to the package insert and directions of the manufacturer and run on a GENE-UP^®^ real-time PCR system (bioMerieux Inc.) with GENE-UP^®^ routine software (version 3.2.0.46).

### 2.2. Screening of E. coli Isolates

Eighty-three strains of EHEC of various serotypes isolated from beef or cattle and cases of human disease (Appendix A) were grown overnight at 37 °C in tryptic soy broth (TBS; Difco, Sparks, MD, USA) and then diluted 1:100 in buffered peptone water (BPW; Difco). The diluted strains were then used in the GENE-UP^®^ lysis and tested with the EH1^®^ (*stx* and *eae*) and PEC^®^ (*espK*/*espV*) assays. One hundred strains of STEC (Appendix A) and 95 strains of atypical EPEC (Appendix A) that had been isolated from beef or cattle sources were similarly prepared and lysed, then tested with the EH1^®^ and PEC^®^ assays. All tests were run without technical replicates unless unexpected results were found, then strains were repeated and further interrogated to resolve discrepancies.

### 2.3. Inoculation Study

MicroTally^®^ manual sampling device (MSD; Fremonta, Fremont, CA, USA) beef trim sampling cloths were inoculated with 10 mL beef purge collected from thawed vacuum-packed beef cuts and trimmings, and with 20 g of 50, 73, or 80% lean ground beef. The ground beef and purge were not prescreened for *stx*, *eae*, *espK*, or *espV*. The percentage lean used on any MSD was arbitrary, as 20 g portions of ground beef were prepared and applied in a rotating manner without order. The purge was spread across an unfolded MSD using a cell spreader (MidSci, St. Louis, MO, USA), then the ground beef was manually spread and pressed into the MSD by a sterilized gloved hand. MSDs were then inoculated with one or two strains of *E. coli* (Appendix A). The strains consisted of 12 EHEC, 12 STEC, and 12 EPEC. Beef-inoculated MSDs were prepared ten at a time using two strains each of STEC, EPEC, and EHEC, with four MSDs inoculated with each STEC and EPEC in combination (Appendix A) to generate samples that could contain *stx*, *eae*, and common O serogroup genes.

For inoculation, a single colony of each strain was taken from the surface of an Oxoid Sorbitol MacConkey agar with 5-bromo-4-chloro-3-indolyl-β-D-galactopyranoside plate (SMAC-BCIG; ThermoFisher Scientific, Waltham, MA, USA) and grown overnight at 37 °C in TSB. The overnight cultures were stored at 4 °C for 24–72 h before diluting in BPW to target ranges of 50 colony-forming units (CFU)/mL for EHEC strains, and 500 CFU/mL for STEC and EPEC strains. CFUs/mL were determined by colony counts observed on tryptic soy agar (Difco) plates that had been incubated overnight at 37 °C. Beef-inoculated MSDs were inoculated with 100 μL of each diluted strain as described, refolded, returned to their Whirl-Pak bag, and held at 4 °C 16 h. Then 200 mL of either prewarmed (42 °C) modified TSB containing casamino acids (mTSBca; Difco) or BPW was added to each MSD. Each MSD was mixed in a laboratory blender (BagMix 400, Interscience, Woburn, MA, USA), 1 mL removed for aerobic plate count, and then incubated for 8 h at 42 °C. After 8 h, MSD bags were opened, a portion was removed for 8 h analysis, and then the remaining MSD sample was split with each portion incubated for an additional 7 or 16 h at 42 °C in a programmable incubator that then chilled the MSD samples at 4 °C. Portions were then removed from these 15 h (8 h + 7 h) and 24 h (8 h + 16 h)-enriched MSD samples for the molecular tests. Each sample in either media and at each time point was treated as an independent data point for analysis. All EH1^®^ and PEC^®^ assay tests were run without technical replicates unless unexpected results were found, then samples were repeated and further interrogated to resolve discrepancies. Through this process, two BPW samples were found to be cross-contaminated with an EHEC and dropped from the analysis.

### 2.4. Aerobic Plate Count (APC)

One milliliter portions of the beef inoculated MSD samples as prepared above were serially diluted (1:10) in BPW and plated to Petrifilm AC (Neogen Corp., Lansing, MI, USA). Petrifilms were incubated at 35 °C for 48 h, and colonies were counted.

### 2.5. Natural Beef Broths

Regulatory beef broths (n = 352) were processed according to the FSIS Microbiological Laboratory Guidebook (MLG) Chapter 5C.03 [11]; they were found to be positive for *stx* and *eae*, and were blinded for identity and archived by FSIS field service laboratories (FSL) as glycerol stocks and then shipped to the United States Meat Animal Research Center (USMARC) for testing. Glycerol stocks were prepared by combining and mixing 30 mL of broth from a *stx*- and *eae*-positive sample with 15 mL 50% sterile glycerol in a 50 mL conical tube. Glycerol stocks were held at −20 °C and periodically shipped on dry ice by overnight courier. Once they arrived in the USMARC laboratory, glycerol stocks were thawed, GENE-UP^®^ DNA lysis prepared, and then they were aliquoted into 1 mL vials to avoid multiple freeze–thaw cycles. The presence of *stx* and *eae* and *espK*/*espV* were tested on all broths using GENE-UP^®^ EH1^®^ and PEC^®^ test kits, respectively. All *stx*- and *eae*-positive broths were further tested for *E. coli* O157:H7 and non-O157 serogroups (O26, O45, O103, O111, O121, and O145) using GENE-UP^®^ ECO^®^ and EH2^®^ test kits, respectively. Broths suspected of containing STEC or EHEC were then taken forward to culture isolation as described below.

### 2.6. Culture Isolation

All broths were cultured for STEC and EHEC. First, a 1 mL glycerol aliquot was thawed, and a 500 μL portion was mixed with 500 μL phosphate-buffered saline (PBS) containing 0.05% polyoxyethylene (20) sorbitan monolaurate (TWEEN-20; PBS-T; MilliporeSigma, Burlington, MA, USA). Then 20 μL of serotype-specific immuno-magnetic separation (IMS) beads were added if suspect O groups (O157 and non-O157) had been identified. Anti-O157 Dyna-beads (ThermoFisher Scientific, Waltham, MA, USA) were used first, then the appropriate non-O157 IMS beads (Romer Labs, Newark, DE, USA) were sequentially used when a sample was found positive for more than one serogroup. The sample–bead mixes were mixed in a rotary plate shaker (400 rpm; Micromixer MX4, FINEPCR, Seoul, Republic of Korea) for 15 min, and then the beads were captured using a Thermo KingFisher IMS robot. If testing indicated more than one serogroup was present, at this point, the next IMS bead was added, and the capture process was repeated as described above. IMS beads were diluted in BPW and spread plated onto plates of Chromagar STEC (DRG International, Springfield, NJ, USA), modified Rainbow agar containing (5 mg/L novobiocin, 0.15 mg/L potassium tellurite, and 0.05 mg/L cefixime; mRBA; BioLog, Hayward, CA, USA) and washed blood agar containing mitomycin C (WBAM; [12]). The enrichment remaining after IMS procedures was streaked for isolation onto Chromagar STEC. If no serogroup was indicated by molecular screening, then the broth was diluted in BPW, and 50 μL was spread plated onto Chromagar STEC (1:50 and 1:250 dilutions), mRBA (1:250 and 1:500 dilutions), and WBAM (1:500 and 1:5000 dilutions). All plates were incubated at 37 °C overnight, and up to twelve colonies of suspect phenotypes (pink on Chromagar STEC, enterohemolytic on WBAM, non-white or non-clear on mRBA), if present, were picked per plate to 96-well blocks of TSB, grown overnight at 37 °C, to be screened by multiplex PCR identifying *stx*_1_, *stx*_2_, *eae* and *ehxA* [13]. Once an isolate was determined to be a STEC or EHEC, it was serogrouped by molecular and serologic methods as previously described [14].

### 2.7. Statistical Analysis

To determine if the addition of the *espK*/*espV* improved the classification of EHEC vs. EPEC, STEC, or a mix of EPEC and STEC, samples were classified into groups “EHEC” or “not EHEC” based on their screening results. For *stx*/*eae* assay alone, samples were classified as “EHEC” if they were *stx*+/*eae*+. Otherwise, they were classified as “not EHEC”. For the *espK*/*espV* assay, samples were classified as “EHEC” if they were *stx*+/*eae*+ and *espK*/*espV*+; all other results were classified as “not EHEC”. For the inoculated sample study, the media used (BPW or mTSBca) or the duration of incubation (8, 15, or 24 h) were tested for any effect on the classification results, with all sample points analyzed as independent measures. For the natural sample study, the type of product (ground, intact, non-intact, or trim) was tested for an effect on the classification ability of the assays. Classification ability was calculated as the Area Under the Receiver Operating Characteristic Curve (AUC of the ROC, hereafter AUC [15]). An AUC value of 50% can be interpreted as “random agreement” between two classifiers (in this case, the prediction of the assay and the true classification), while an AUC value of 100% would be perfect agreement; an AUC of 70% or more is generally considered “good agreement” [16]). AUCs were compared using planned-comparison DeLong’s tests [15] with a family-wise alpha of 0.05 and a Bonferroni correction [17]. All analyses were performed in the R Statistical Programming Language (r, [18]) using the packages pROC [19] and caret [20].

## 3. Results and Discussion

### 3.1. Examination of E. coli Strains for espK/espV

Three groups of *E. coli* were examined for the pathogenicity markers *espK* and *espV*. All EHEC (n = 83) strains contained either or both markers (Appendix A), while all (n = 100) STEC strains lacked *espK* and *espV* (Appendix A). EPEC strains, however, provided mixed results with 60% of the 95 isolates screening positive for one or both of the markers (Appendix A). The specificity of *espK*/*espV* for EHEC over generic *E. coli* was previously shown using bioinformatics [21]. There, the presence of *stx* and *eae*, and *espK* and *espV* were determined in 3901 *E. coli* genomes, of which 775 were EHEC and 3126 were nonpathogenic lacking *stx* and *eae*. Performing in silico PCR using the GENE-UP^®^ primer sequences for *stx*, *eae*, *espV*, and *espK* on the genomes found 99% (774/775) of EHEC genomes possessed *espK* and/or *espV*. Further, *espK* and *espV* were not identified in any of the nonpathogenic *E. coli* genomes. An unpublished portion of that analysis identified 173 other EPEC genomes where 122 (70.5%) possessed *espK* and/or *espV* (V. Dutta, personal communication). Similar to the results here, Delannoy et al. [7] showed EHEC could be discriminated from non-EHEC strains based on the presence of various combinations of effector genes. Out of the effectors examined among 340 EHEC, 392 EPEC, 193 STEC, and 175 apathogenic *E. coli* strains, *espK* and/or *espV* were more prevalent in EHEC (98.5%) than in EPEC (54.1%) [7]. When fifty-five reference EHEC strains were examined, all but one, an EHEC-O26:H11, possessed *espK* and/or *espV* [10]. Further examination of 195 EHEC-O26:H11 showed that one lacked *espK* while three lacked *espV* [22]. The absence of these two markers is exceedingly rare in EHEC.

The presence of *espK* and *espV* in EPEC may reveal their relationship to EHEC. EPEC may be typical or atypical (aEPEC) and differ in several characteristics. Typical EPEC are a leading cause of diarrhea in infants in developing countries, whereas aEPEC are rare but a more important cause of diarrhea in industrialized countries [23]. The primary genetic difference is that aEPEC lack the plasmid-encoded bundle-forming pilus operon [1]. The most commonly reported serogroups of aEPEC isolated from humans are O26, O55, O86, O111, O119, O125 and O128 [23]. The collection of EPEC strains used here was aEPEC, and of these common serogroups, only O26 was well represented. This may be the result of these strains being sourced only from beef and/or the influence of media bias during culture isolation. EPEC-O26 strains such as these that are positive for *espK* and *espV* have been termed EHEC-like due to the presence of *stx*-phage remnants in their genomes [24]. There were twenty-one total O26-EPEC in the collection, six of which possessed *espK* and/or *espV*. Other EPEC strains of serogroups O45, O103, and O145 are possibly EHEC-like as well. All three of the EPEC-O45 and all fifteen of the EPEC-O145 isolates tested possessed *espK* and/or *espV*, while five of the six EPEC-O103 possessed *espK* and/or *espV*. Despite these EHEC-like strains, *espK* and *espV* can still serve as good or better molecular targets than *eae* for identifying EHEC and helping resolve contamination by EHEC from mixed cultures of STEC and some EPEC.

### 3.2. Distinguishing Beef Samples Inoculated with EHEC from Those Inoculated with STEC and EPEC

Inoculated beef MSDs were enriched in mTSBca and BPW and then tested after 8, 15, and 24 h of incubation for *stx*, *eae*, and *espK*/*espV*. For analysis purposes, the inoculated samples were classified as either “EHEC” or “not EHEC” based on the results of the *stx*/*eae* screen and the *stx*/*eae*-*espK*/*espV*-combined screen. For the *stx*/*eae* screen, samples were classified as “EHEC” when inoculated with an EHEC or a mixture of STEC and EPEC and “not EHEC” when inoculated with either a STEC or an EPEC. Then, when the *espK*/*espV* assay was included in the classification, the samples inoculated with an EHEC or a mixture of STEC and an EPEC known to possess *espK* and/or *espV* were classified as “EHEC”, otherwise mixtures of STEC and EPEC lacking *espK* and/or *espV* were classified as “not EHEC” along with the samples inoculated with only a STEC or an EPEC.

Thirty-six of the inoculated beef MSDs in each set (mTSBca and BPW) were expected to screen positive for *stx*/*eae,* and 26 positives with the combined *stx*/*eae*-*espK*/*espV* screen could be classified as “EHEC” for analysis. Actual numbers positive for *stx*/*eae* screening ranged from 34 to 40 depending on media and time point, and 26 to 31 for the combined *stx*/*eae*-*espK*/*espV* screen. In all cases, the combined *stx*/*eae*-*espK*/*espV* screen narrowed the number of potential positive samples by 9 to 11 at the 8 and 15 h time points and 5 to 6 at the 24 h time point. These variations are likely due to the natural occurrence of STEC and aEPEC in the ground beef products used to prepare the inoculated MSDs. Background prevalence of *stx* in ground beef has been shown to be as high as 24.1%, while intimin positivity was 20.5% [25] and ranges from 5 to 50% depending on the sampling device used [26]. Considering the prevalence observed of *espK* and/or *espV* among aEPEC strains, the presence of background levels of these two markers is not unexpected.

The AUC of the *stx*/*eae* only assay was 68.0% (95% confidence interval [CI] 64.4–73%) and significantly lower than that of the *stx*/*eae*-*espK*/*espV* combined screen whose AUC was 76.8% (95% CI 72.25–81.15%; Z = −6.03; *p* = 1.679 × 10^−9^, Figure 1). The AUC depicts the tradeoff between the true positive rate and the false positive rate of different classifiers “EHEC” and “not EHEC”. The increase in AUC from 68.0% to 76.8% signifies an increase in agreement to good agreement with the addition of *espK*/*espV* to the screening assay. No significant differences between media types or durations of incubation were found (all *p* > 0.05; Appendix A). Thus, using *espK* and *espV* should increase the specificity of identifying higher-risk materials and allow a more rapid release of materials that do not contain an EHEC.

### 3.3. Analysis of Suspect Beef Samples

Beef samples that had been originally screened as positive for *stx* and *eae* (n = 361) by FSIS FSLs were re-examined for *stx*, *eae*, and then *espK*/*espV*. After frozen storage and shipment to USMARC, 319 were found positive for *stx* and *eae,* while 42 lacked one or both of the markers. The pathogenic *E. coli* markers *espK*/*espV* were detected in 56% (178/319) of the *stx*/*eae*-positive broths and 14% (6/42) of the *stx*/*eae*-negative broths. Upon closer examination, five of those six broths were positive for *stx* (n = 2) or *eae* (n = 3). These broths likely lost either the *stx* or *eae* signal of the initial FSIS FSL screening test during the dilution in glycerol and/or freezing–thawing process. The reason for the presence of *espK*/*espV* in the single *stx*- and *eae*-lacking broth could be attributed to the same processes of dilution, freezing, and thawing since the initial broth was identified as having *stx* and *eae*. The initial screening by the FSIS FSLs was performed according to the 5C.03 [11] procedure and used an alternate real-time PCR test that the one used here may have been unable to replicate 100%.

Studies have shown the freeze–thaw process can bias bacterial detection and recovery. Freezing and thawing were shown to destroy Flavobacterium cells so that DNA became undetectable by PCR [27]. A loss of 74% in cell viability reported in sludge samples was attributed to the physical changes that take place during freezing and thawing [28]. Of 272 pig cecal samples tested before and after a 6-month freezing period the number with colistin-resistant isolates was higher in fresh samples (76%) compared to frozen ones (20%; [29]). These and other studies have demonstrated that the freeze–thaw process can decrease bacterial diversity in the samples [28,29,30]. Since the regulatory beef enrichments examined here are mixed cultures of STEC, EPEC, and other microorganisms, it is not unexpected that the freeze–thaw process reduced this diversity, impacting molecular detection as well as subsequent culture isolation results. Nevertheless, the most efficient way to perform this study required the glycerol freezer storage of samples identified by the FSIS FSLs and was unavoidable.

All 361 suspect broths were cultured for EHEC, which resulted in 42 EHEC, 82 STEC, and 67 EPEC isolates from 146 samples (Appendix A). Most (n = 28) samples that yielded an EHEC only had an EHEC isolated, but six samples also had a STEC, three an EPEC, and three others both a STEC and an EPEC present with the EHEC. Three samples were found to have two different EHEC present. Forty-four other samples contained a STEC, 34 an EPEC, and 27 a STEC and an EPEC. Thus, the screening results for *stx* and *eae* were confirmed by culture in 67 samples (containing an EHEC or a STEC with an EPEC) and partially confirmed in 79 (Figure 2). Amongst the EHEC that could be serogrouped were seven O157:H7, ten EHEC-O103, and six EHEC-O26. In addition, less common EHEC were isolated, including five EHEC-O182, four EHEC-O177, and one EHEC-O5. Nine additional EHEC were untypable (Ount) using the typing sera available. Three of the EHEC-Ount occurred in samples with a common EHEC (one each: EHEC-O26, -O103, and -O157).

When the presence of *stx*, *eae*, and *espK*/*espV* are considered with the results of culture isolation (Figure 2), only one STEC was recovered from those samples that were lacking *stx* and *eae*. Whereas 43 samples that contained *stx* and *eae*, but lacked *espK*/*espV* (n = 141) were found to contain a STEC and/or an EPEC. This group of samples was also the source of one EHEC of an uncommon serogroup. This EHEC possessed the *espK*/*espV* genes, so the discrepancy between PCR and isolation may be attributed to the impacts of dilution and freeze–thaw of the primary sample as discussed above. Otherwise, 97% (38/39) of the EHEC isolates were recovered from samples that contained *stx*, *eae*, and *espK*/*espV*.

One hundred and forty samples were *stx*-, *eae*-, and *espK*/*espV*-positive but not found to have an EHEC present. Forty-one of those, though, contained an EPEC, (22 being of a common non-O157 serogroup O26, O45, O103, and O145) as the likely source of the *espK*/*espV*. Still, 78 samples did not yield any isolates, and 46 did not yield isolates to explain the positive screening results. Culture bias likely plays a role in the variability of the results observed. Both the enrichment broth and the plating media can affect the detection and recovery of various STEC strains due to competition during growth [31]; therefore, it is important that multiple approaches for isolation of STEC be taken [32]. The beef enrichment broths provided by the FSLs used mTSBca, and not all STEC grow equally well in broth media based partly on the sample type. For instance, modified TSB was shown to have advantages compared to TSB or *E. coli* (EC) broth when enriching STEC from cattle feces [33], and TSB was recommended as the STEC enrichment broth for clinical stool specimens instead of gram-negative or MacConkey broths [34]. Differences in the growth of STEC strains in the enrichment are then influenced again by the plating media used to isolate them. Isolation here used mRBA, Chromagar STEC, and WBAM because this combination of agars provides ample opportunities to identify colonies of STEC [14]; however, other media types such as Posse Agar, Tryptone Bile X-glucuronide agar, Rapid *E. coli* O157:H7 agar, and MacConkey agar have been shown to support the growth of strains other media may not [35,36].

The use of *espK*/*espV* as an additional screening test distinguishes beef at risk of being contaminated by a potential EHEC from that contaminated by mixed cultures of STEC and EPEC, thus allowing a more focused group of samples to be further interrogated for culture confirmation. Results here suggest using *espK*/*espV* can reduce the number of samples by about half, but some meat products may be more suited to the utility of the *espK*/*espV* screen than others. The FSL samples represented beef trim (n = 153), ground beef (n = 151), intact beef (n = 38), and non-intact beef (n = 19). Analyzing the results of the *stx*/*eae* and *espK*/*espV* screens according to the culture results and applying the “EHEC” and “not EHEC” classifiers, the AUC of the *stx*/*eae* only assay was just 50.5% (95% CI 49.6–51.4%) or equal to random agreement (Figure 3; Appendix A). Using the *stx*/*eae*-*espK*/*espV*-combined screen whose AUC was 69.8% (95% CI 64.4–75.1%) provided a significantly better classification ability (Z = −7.11, *p* = 1.17 × 10^−12^, Figure 3). Sample product type was not found to be a significant predictor of classification ability for the *stx*/*eae* assay (all *p* < 0.05), but for the *stx*/*eae*-*espK*/*espV* screen, non-intact product type samples have significantly poorer classification ability compared to all others (AUC for non-intact product types = 50%, all *p* < 0.05), likely influenced by the low sample size (Figure 3; Appendix A).

The benefits of the *stx*/*eae*-*espK*/*espV* testing approach have been described for French beef and dairy samples. The discriminatory power of EHEC screening was similar to that observed here. When 1739 beef enrichments were tested, 180 were *stx*/*eae*-positive, while 90 (50%) were *stx*/*eae*-*espK*/*espV*-positive. It was concluded that screening for *stx*, *eae*, *espK*, and *espV* was a better approach to narrow down EHEC screening of beef enrichments compared to methods that relied on *stx*, *eae*, and O-group genes [9]. In a study that examined dairy products, for PCR analysis of 1451 milk and raw milk cheeses positive for *stx*/*eae*, the addition of *espK*/*espV* resulted in better selectivity and a reduction of the number of presumptive positive samples. The impact of *espK*/*espV* on reducing the number of positive samples depended on the animal species. Reductions were 26% in milk and cheese samples from goats, 29% in milk and cheese from cows, and 52% in milk and cheese from sheep [37]. In addition, *stx*/*eae*-*espK*/*espV* testing identified samples contaminated by an uncommon EHEC-O80 as well as common EHEC serogroups [37].

## 4. Conclusions

The pathogenic *E. coli* screening assay for *espK*/*espV* was shown to be useful and approximately double the specificity of EHEC screening. Using a *stx*/*eae*-*espK*/*espV* screening approach readily identified natural samples contaminated by EHEC of common and uncommon serogroups. In their report on STEC, the Food and Agriculture Organization of the United Nations proposes that STEC strains of significance be categorized on virulence gene content rather than their serogroup or serotype with strains causing disease that progress to hemolytic uremic syndrome and bloody diarrhea be the primary focus of public health agencies [38]. Of the strains they describe, all are EHEC possessing *eae* and either *stx*_2a_, *stx*_2c_, or *stx*_1a,_ as well as those that only express *stx*_2d_. The use of *espK*/*espV* with *stx*/*eae* can help direct testing to meet these goals, as contamination by all EHEC except those expressing *stx*_2d_ alone can be rapidly identified. The use of *espK*/*espV* as an initial screen without *stx*/*eae* is also a possibility needing further investigation. It is an acceptable approach [39], but the background occurrence of EPEC carrying these markers needs further evaluation.

## Figures and Tables

**Figure 1 foods-14-00382-f001:**
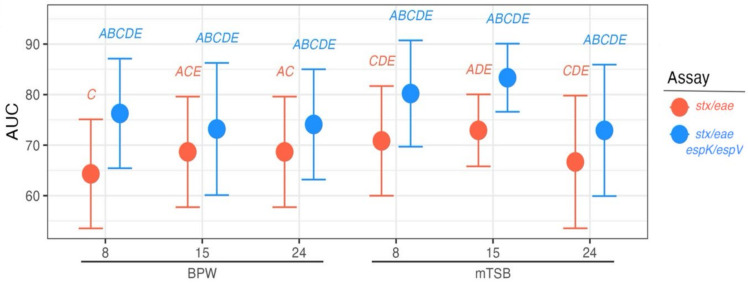
Ability of STEC screening assays without and with *espK*/*espV* to correctly classify EHEC or non-EHEC inoculated beef MSD samples across timepoints and media types. Points represent the AUC of the assay with 95% confidence intervals. Letters give significantly different (*p* < 0.05) groups.

**Figure 2 foods-14-00382-f002:**
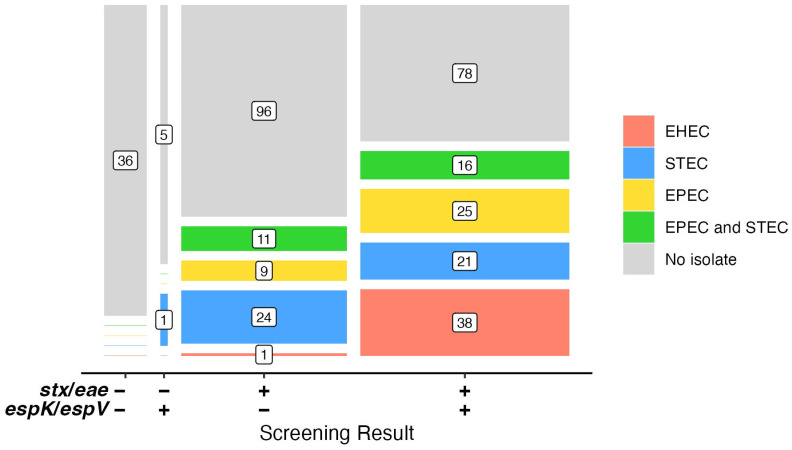
Numbers of EHEC, STEC, and EPEC recovered from beef regulatory broths by category of screening assays *stx*/*eae* (+/−) and *espK*/*espV* (+/−). “Negative” are broths with no isolate recovered. “EPEC and STEC” are broths with both types of strains isolated. Not shown are broths where other strain types were recovered with an EHEC.

**Figure 3 foods-14-00382-f003:**
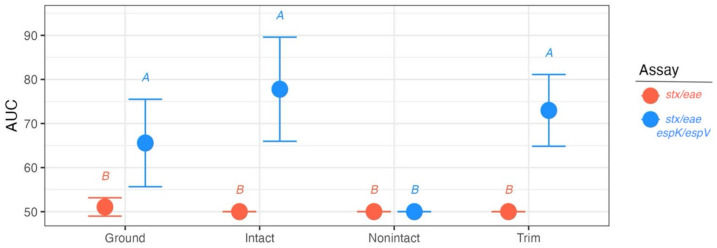
The ability of STEC screening assays without and with *espK*/*espV* to correctly classify EHEC or non-EHEC among beef enrichments according to recovery of EHEC, STEC, and EPEC strains during culture confirmation. Points represent the AUC of the assay with 95% confidence intervals. Letters give significantly different (*p* < 0.05) groups.

## Data Availability

All data and test results are available in the Appendix A.

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
