# Peer review of "Using Pathogenic Escherichia coli Type III Secreted Effectors espK and espV as Markers to Reduce the Risk of Potentially Enterohemorrhagic Shiga Toxin-Producing Escherichia coli in Beef"

_foods, 2025, doi:10.3390/foods14030382_

Round 1
Reviewer 1 Report
Comments and Suggestions for Authors
The manuscript foods-3412752 entitled “Using Pathogenic Escherichia coli Markers espK and espV to Reduce the Risk of Shiga Toxin-producing E. coli in Beef” reports the results on the evaluation of the commercial assay for Pathogenic E. coli (PEC) that targets espK and espV in combination with the screening of the stx and eae genes in order to determine its impact on rising the specificity of enterohemorrhagic E. coli (EHEC) detection. The authors tested the PEC on inoculated beef with a dozen of strains of the EHEC, EPEC and STEC groups. The effect of beef enrichment on the EHEC detection was evaluated using mTSB and BPW. The authors clearly described the methodology and adequately justified the approach. The results are plainly shown, and the discussion is correctly driven. The conclusions are supported by the results and the future perspectives are well established.
The manuscript requires just minor alterations that are indicated below.
Line 57-59. Indicate the origin of the isolates from reference 7.
Line 62. change to E. coli and in italic.
Line 74-75. The proper reference for this sentence is required.
Line 125-126. change was incubated an additional to was incubated for an additional
Line 128. for testing what?
Line 131. change to Petrifilms.
Line 135. change to stocks and then
Line 153. change to 15 min
Line 203. apathogenic ? non-pathogenic, correct?
Line 274-275. This is too speculative. Rewrite.
Line 370. eliminate the double in

Author Response
REVIEWER 1:
The manuscript foods-3412752 entitled “Using Pathogenic Escherichia coli Markers espK and espV to Reduce the Risk of Shiga Toxin-producing E. coli in Beef” reports the results on the evaluation of the commercial assay for Pathogenic E. coli (PEC) that targets espK and espV in combination with the screening of the stx and eae genes in order to determine its impact on rising the specificity of enterohemorrhagic E. coli (EHEC) detection. The authors tested the PEC on inoculated beef with a dozen of strains of the EHEC, EPEC and STEC groups. The effect of beef enrichment on the EHEC detection was evaluated using mTSB and BPW. The authors clearly described the methodology and adequately justified the approach. The results are plainly shown, and the discussion is correctly driven. The conclusions are supported by the results and the future perspectives are well established.
The manuscript requires just minor alterations that are indicated below.
Thank you for your time and effort spent reviewing our paper. I believe that each comment (that I understood) was properly addressed. One exception below, you’ll see.
Line 57-59. Indicate the origin of the isolates from reference 7.
These isolates were from humans, animals, and food sources. To track them, they were from earlier papers by the same group, actually ref 7 cites the strains to 3 previous papers. And one of those papers refers to isolates from 5 previous papers. Since I was only tracking and introducing the data on espK and espV, I did not go into the additional background of the strains and only referred to the espK/V paper. I have kept that the same but expanded the description to explain sources as humans, animals, and food sources. I hope that was what the reviewer was asking for.
Line 62. change to E. coli and in italic.
Got it.
Line 74-75. The proper reference for this sentence is required.
That is a statement from Emilio Esteban of FSIS describing their situation a few years back. Current conversations with staff there confirm this is still the case. I have cited the Federal Register notice where FSIS describes the situation as false positive rates in background when posting the expansion of STEC testing of beef products.
Line 125-126. change was incubated an additional to was incubated for an additional
Done.
Line 128. for testing what?
Changed to “molecular tests” and added that these were the EH1® and PEC® assays.
Line 131. change to Petrifilms.
I’ve always considered Petrifilm to be one of those words that is its own plural. An irregular plural, not really a plural tantum, I think. Anyway, I’ll give this to the reviewer and let the Copy Ed sort it out,
Line 135. change to stocks and then
Done.
Line 153. change to 15 min
Done.
Line 203. apathogenic ? non-pathogenic, correct?
Not sure, line numbers are a little off. If referring to use of “nonpathogen E. coli genomes” I don’t think that needs changing. If referring to the next lines describing the EPECs with espK/V the text has established that EPEC are non-pathogens when compared to EHEC. So again, I do not see the need to add extra description to them. No changes made, but happy to address if I misunderstood the request.
Line 274-275. This is too speculative. Rewrite.
OK, so loosing either stx or eae is an acceptable possibility, but losing both is a further leap, I get that. I should add (only here not in text, as it would become too confusing and go beyond scope of this paper) that using alternate stx/eae assays show varying presence absence of stx and eae in these samples. I’ve revised this to focus only on the single sample described with espK/espV in the absence of stx/eae. The original sentence referred to samples that were stx, eae, and espK/espV negative. This section of text now reads better (at least to me). I hope it meets reviewer’s wishes.
Line 370. eliminate the double in
Fixed. It always surprises me how little things like this get missed considering the multiple levels of proofreading a manuscript gets.
Reviewer 2 Report
Comments and Suggestions for Authors
Thank you for having an opportunity to review the manuscript entitled "Using Pathogenic Escherichia coli Type III Secreted Effectors espK and espV as Markers to Reduce the Risk of Shiga Toxin producing Escherichia coli in Beef ".
This is a study that analyzed the applicability of testing additional virulence factors (apart from stx1, stx2, eae, and O-specific) associated with the risk of contracting EPEC/EHEC/STEC foodborne illness.
The introduction provides adequate information regarding the present status of laboratory practices in testing for EHEC. The text is articulated comprehensively and effectively delineates the probable deficiencies in present regulatory monitoring while proposing alternative solutions to enhance understanding of the transmission of pathogenic EC.
In Line 85, (but also 391) I am uncertain whether the term "specificity" should be substituted with "sensitivity" within the context of the study?
I have concerns concerning the title of the manuscript. The title emphasizes STEC; however, the Introduction suggests that the inclusion of espK and espV pertains more specifically to EHEC. The overarching concept appears to be stx1+, stx2+, eae+, espK, and espV+ EHEC, rather than STEC, as the latter does not necessarily include eae. The authors, actually indicated this in Lines 239-241. Could the authors, please, clarify this?
The Materials and Methods section adequately describes the experimental procedures of the study, detailing the commercial kits utilized for the molecular screening of the EC DNA for virulence and O-specific components as well as sampling device manipulation. Kindly observe that line 127 lacks information regarding how the authors evaluated these values (50, specifically 500 cfu/mL)?
The Results and Discussion sections are articulated effectively, delineating specific limitations of the study. Notably, the loss of factors from field samples subjected to freeze/thaw cycling is addressed, alongside limitations arising from unmonitored beef purges. In Line 361, it would be beneficial for the authors to elaborate on or rephrase the findings that are further detailed in Table S4. The circumstances surrounding the resuscitation of the 361 samples remain ambiguous, particularly regarding the observation that only approximately one third were subsequently confirmed through cultural methods.
Author Response
REVIEWER 2:
Thank you for having an opportunity to review the manuscript entitled "Using Pathogenic Escherichia coli Type III Secreted Effectors espK and espV as Markers to Reduce the Risk of Shiga Toxin producing Escherichia coli in Beef ".
This is a study that analyzed the applicability of testing additional virulence factors (apart from stx1, stx2, eae, and O-specific) associated with the risk of contracting EPEC/EHEC/STEC foodborne illness.
The introduction provides adequate information regarding the present status of laboratory practices in testing for EHEC. The text is articulated comprehensively and effectively delineates the probable deficiencies in present regulatory monitoring while proposing alternative solutions to enhance understanding of the transmission of pathogenic EC.
Thank you for your time and effort reviewing our manuscript.
In Line 85, (but also 391) I am uncertain whether the term "specificity" should be substituted with "sensitivity" within the context of the study?
As I understand the terms (and double checked with Google) “sensitivity” of the test refers to its ability to identify samples that contain an EHEC. So, a more sensitive test has fewer false negatives (misses fewer positive samples). Whereas “specificity” of the test describes its ability to correctly identify samples containing an EHEC. More specific tests have fewer false positives. So, in the context of stx/eae testing, with and without espK/V testing, adding espK/V increases specificity and reduces the number of potential positive samples. For clarity, at Line 85 when “specificity” is used, “(i.e. fewer false positives)” follows to define what we are meaning in the study.
I have concerns concerning the title of the manuscript. The title emphasizes STEC; however, the Introduction suggests that the inclusion of espK and espV pertains more specifically to EHEC. The overarching concept appears to be stx1+, stx2+, eae+, espK, and espV+ EHEC, rather than STEC, as the latter does not necessarily include eae. The authors, actually indicated this in Lines 239-241. Could the authors, please, clarify this?
I agree. STEC vs EHEC terminology often get hit as improper depending on the reader/reviewer’s point of view or priorities. For some EHEC are STEC and “EHEC” should not be used because it only describes the disease caused by a subset of strains that possess stx and should be called STEC. I prefer to set EHEC apart from STEC. For clarity I have added “potentially enterohemorrhagic” to the title.
The Materials and Methods section adequately describes the experimental procedures of the study, detailing the commercial kits utilized for the molecular screening of the EC DNA for virulence and O-specific components as well as sampling device manipulation. Kindly observe that line 127 lacks information regarding how the authors evaluated these values (50, specifically 500 cfu/mL)?
Ah, I see that. OK, I have added a quick explanation that the CFUs were determined by plating on TSA and colony counting. The actual numbers are provided in the Supplemental Tables S3a and S3b where you can see how good or poorly we were at hitting our targeted level. I feel successful if I’m within a log of the target, which these were.
The Results and Discussion sections are articulated effectively, delineating specific limitations of the study. Notably, the loss of factors from field samples subjected to freeze/thaw cycling is addressed, alongside limitations arising from unmonitored beef purges. In Line 361, it would be beneficial for the authors to elaborate on or rephrase the findings that are further detailed in Table S4. The circumstances surrounding the resuscitation of the 361 samples remain ambiguous, particularly regarding the observation that only approximately one third were subsequently confirmed through cultural methods.
I’m not quite sure what the ask is here. But I have changed the spin at line 361, from one of removing false positives to one of taking forward fewer samples that are more likely to contain an EHEC.
Table S4 provides the raw results of the screening and culture of the natural broths. Where the impact of the espK/V assay can be seen to correlate with the isolation of EHEC of a number of serogroups. I have simplified the table removing a couple redundant columns.
The Methods section describes the pains taken to isolate any potential STEC, EPEC, or EHEC from these samples. For brevity some of the details were left out and readers referred to a more in-depth description in an earlier paper. Due to limited time and resources exhaustive culture work was not performed. Rather, up to 12 colonies per plate type and process (IMS target) were examined where colonies interrogated per sample ranged from 20 to 144. Often many plates did not provide many suspect colonies to examine. It is not uncommon (even with fresh enrichments that have never been frozen) to have non-confirmable STEC/EHEC screening tests. That is, after all is the point of the PEC test; to reduce this number, which I think we demonstrate. Lastly, as mentioned to Reviewer 1, I have more information on these broths than revealed in this paper that focused only on using GENEUP assays (EH1, EH2, ECO and PEC). A similar paper is in preparation focusing on the iQCheck test kits and droplet PCR. From that additional data, I am confident that the “true” positives withing the natural broth set were identified by culture work. The Methods have been revised to add that up to 12 colonies were examined per plate.
Reviewer 3 Report
Comments and Suggestions for Authors
The manuscript by Joseph M. Bosilevac and colleagues, titled "Using Pathogenic Escherichia coli Type III Secreted Effectors espK and espV as Markers to Reduce the Risk of Shiga Toxin-producing Escherichia coli in Beef" focuses on the study of genetic markers for distinguishing pathogenic lines of Escherichia coli, specifically EHEC from others. The manuscript is quite interesting and will undoubtedly be applicable in diagnostic practice. However, to accept the manuscript, the following comments need to be addressed:
I will provide line numbers according to the latest (second, as I understand) version of the manuscript.
Lines 9-11: What are the differences between the second and third affiliations? What does MO 3 and MO 2 mean? It really matters?
Line 66: Typo "coil" instead of "coli"
Line 102: The Table does not clarify the difference between human and feces. As I understand, are feces obtained from humans?
Line 103: Typo "TBS" instead of "TSB"
Line 121: Which Table S3 is referred to - table S3a or table S3b?
Line 212: How did you obtain the sequences of the primers? Are they included in the instructions for the commercial kit? Or did you participate in its development? It would be advisable to include them in the manuscript, for example, in an supplementary table.
Line 214: I recommend replacing "nonpathogen" with "nonpathogenic"
Figures should be inserted after their mention in the text, not before.
Line 280: Tables should be referenced in the text in order. You have S5 mentioned before S4 (line 314).
Line 309 and Figure 2: Genes should be italicized.
Line 408: Forgot to put as "Table 3b".
Format the References according to the journal requirements and correct authors with some symbols, such as the square root etc. in references # 4, 6, 18, 26, 28, 29.
Author Response
REVIEWER 3:
The manuscript by Joseph M. Bosilevac and colleagues, titled "Using Pathogenic Escherichia coli Type III Secreted Effectors espK and espV as Markers to Reduce the Risk of Shiga Toxin-producing Escherichia coli in Beef" focuses on the study of genetic markers for distinguishing pathogenic lines of Escherichia coli, specifically EHEC from others. The manuscript is quite interesting and will undoubtedly be applicable in diagnostic practice. However, to accept the manuscript, the following comments need to be addressed:
I will provide line numbers according to the latest (second, as I understand) version of the manuscript.
Thank you for your time in reviewing our manuscript and favorable replies.
Lines 9-11: What are the differences between the second and third affiliations? What does MO 3 and MO 2 mean? It really matters?
I see what you mean a cut paste error of some sort was overlooked. There were 2 FSIS laboratories involved, the Midwest and Eastern locations. One in St Louis, Missouri (MO) and the other in Athens, Georgia (GA). Numbers 2 and 3 are supposed to match the authors to the affiliations. I have added the Laboratory Names, corrected the address City, State, and affiliation designators.
Line 66: Typo "coil" instead of "coli"
fixed.
Line 102: The Table does not clarify the difference between human and feces. As I understand, are feces obtained from humans?
My bad. In my previous clinical lab world, humans have “stool” and animals “feces” as defecation samples. All uses of “feces” in Table S1a refer to strains recovered from cattle feces or a RAM swab. The foot notes at the bottom of the table I thought adequately defined this. But since these were provided as supplemental tables, I did not as rigorously set them up for clarity. Many were cut/paste straight from a lab book. At line 102 I have clarified the EHEC were from beef or cattle and cases of human disease. In table S1a I have added asterisks to call reader attention to the definitions at bottom of the Table (same for S1b and S1c too).
Line 103: Typo "TBS" instead of "TSB"
fixed.
Line 121: Which Table S3 is referred to - table S3a or table S3b?
Nice catch, in an earlier version S3a and S3b were combined but then later split by media type. Corrected to “Tables S3a and S3b” in text.
Line 212: How did you obtain the sequences of the primers? Are they included in the instructions for the commercial kit? Or did you participate in its development? It would be advisable to include them in the manuscript, for example, in an supplementary table.
I don’t think that will happen. The primers are part of a kit and the manufacturer doesn’t want to share those. This information at Line 212 was part of an IAFP presentation I did in collaboration with the R&D team at bioMérieux . Their bioinformatician did the work with the in silico primer sequences and I was never privy to them. And if they were shared with me that is probably covered by an NDA. However, there is the Fach et al Patten Application (US 2015/0176064 A1) that lists many primers. I am not sure if bioMérieux used these directly or had to modify them to make the commercial kit.
Line 214: I recommend replacing "nonpathogen" with "nonpathogenic"
I have fond 2 instances of “nonpathogen" in the paper (line 214 and 216), and have replaced both with "nonpathogenic". I agree it is better word use.
Figures should be inserted after their mention in the text, not before.
I’m not a type setter and was trying to position them where they did not break the flow of text, thinking the Journal Copy Ed would be the final say. They’ve been slid around to fit the pages now and occur after their call out.
Line 280: Tables should be referenced in the text in order. You have S5 mentioned before S4 (line 314).
OK. The stats tables were separate and added last. And as I mentioned, they were Supplemental, so I wasn’t treating them as seriously as a table callout in the text. All fixed.
Line 309 and Figure 2: Genes should be italicized.
got it, done.
Line 408: Forgot to put as "Table 3b".
fixed
Format the References according to the journal requirements and correct authors with some symbols, such as the square root etc. in references # 4, 6, 18, 26, 28, 29.
Eww, that’s bad. I did not see those. Apparently there was some sort of MSWord cut paste font craziness afoot when filling in the MDPI Foods template from a plaintext doc. All the oddities have been removed (turned out all be accented characters in names) and the ref’s cross checked for accuracy.